# Let It Out (LIO) study: protocol for a mixed-methods study to optimise the design and assess the feasibility of an online emotional disclosure-based intervention in UK hospices

Daisy McInnerney 🄳 , Bridget Candy, Patrick Stone, Nuriye Kupeli

Marie Curie Palliative Care Research Department, Division of Psychiatry, UCL, London, UK

**Correspondence to**
Daisy McInnerney;
daisy.mcinnerney.18@ucl.ac.uk

## ABSTRACT

**Introduction**  The current COVID-19 pandemic has forced hospices to look for more ways to support people remotely, including psychological support. Emotional disclosure-based interventions hold potential as a way of providing support remotely. However, evidence of their efficacy in people with terminal illness is mixed. Reviews have highlighted this may be due to interventions not being tailored to the unique needs of this population. In response to this, we are developing Let It Out (LIO), an online, self-guided emotional disclosure-based intervention tailored for people living with terminal illness.

**Aims**  The primary objective of the study is to optimise the design of the LIO intervention. Secondary objectives include assessing its acceptability and feasibility; exploring potential impact on well-being; identifying potential adverse effects; and informing choice of outcome measures for potential future evaluation.

**Methods and analysis**  A single arm, mixed-methods, multisite, longitudinal study. Up to 40 people living with a terminal illness under the care of hospices in England and Scotland will receive the online LIO intervention. LIO consists of 3, self-guided expression sessions over 2 weeks. The primary outcome measures are (1) a structured feedback form completed by participants after the final expression session; and (2) semi-structured interviews and focus groups with ≤15 patient participants, ≤30 hospice staff and ≤15 informal carers. These quantitative and qualitative data will be triangulated via process evaluation to inform optimisation of the intervention design. Secondary outcome measures include validated measures of physical and psychological health collected at baseline and after the final expression session (immediately, 1, 4 and 8 weeks after); and data on recruitment, retention and fidelity.

**Ethics and dissemination**  The study is approved by the University College London Research Ethics Committee (reference: 15281/002). The findings will be shared through peer-reviewed scientific journals and conferences, and traditional, online and social media platforms.

## INTRODUCTION

People living with terminal illness carry a unique emotional burden and can experience

## Strengths and limitations of this study

► The Let It Out intervention has been developed based on evidence from a scoping review of previous research, and refined drawing on insights from testing it in a slightly different form in another population and feedback from key stakeholders.

► The acceptability and feasibility of the intervention will be explored using a rich, mixed-methods approach, using process evaluation to triangulate quantitative recruitment, retention and acceptability data with qualitative written and interview feedback.

► Recruitment may be restricted by the relatively short recruitment period and potential limitations on hospice staff capacity (due to COVID-19 related pressures on hospice teams).

► The study is taking place during a time of transition and ongoing change in methods of care, necessitated by the evolving COVID-19 pandemic that may restrict generalisability of the findings outside of this context.

► This is a single-arm study not designed to formally evaluate efficacy of the intervention.

significant distress.[1–4] Psychological support is a key part of the holistic care provided by hospices and other palliative care services.[5–8] But evidence suggests psychological service provision at the end-of-life is currently inadequate, and services are often under-resourced and overburdened.[9–11] Since the start of the COVID-19 pandemic in early 2020, hospice funding and capacity have been placed under even more threat.[12] Hospices are increasingly turning to telehealth approaches to deliver care to patients.[13] At present this is necessary to help minimise face-to-face contact in line with recommended social distancing practices that are particularly crucial for this vulnerable population.[14] However, delivery of care using such methods is potentially cost-effective and time-effective, and if found to

be beneficial, may endure beyond the resolution of the pandemic to help broaden access to palliative care.

Emotional disclosure (ED)-based therapies, such as expressive writing or talking, are easy to implement as a means of support, with minimal requirement for staff supervision and potential to be delivered online.[15 16] The traditional expressive writing protocol is based on the principle that expressing feelings about personal, traumatic experiences can provide comfort.[17] Indeed, there is evidence that such interventions can provide psychological and physical benefits in healthy populations.[18] Expressive writing has also been adapted, to include, for example, disclosing about positive events,[15 19 20] writing or talking about stress from a compassionate stance[21–23] or writing about future goals.[24 25] Writing from a compassionate stance has been shown to reduce negative affect[22 26] and improve mood.[27] However, evidence of benefit of ED-based interventions in palliative populations is more mixed.[28 29] The mixed results may be because there has been a lack of research into developing an ED-based intervention tailored to the specific needs of this population.[28] The underlying causal mechanisms, and thus the outcome measures that may be most suitable to capture any potential benefit of such an intervention in this population, also remain unclear.

Based on this gap in the literature, our group conducted a scoping review of ED-based interventions in people with terminal illness.[29 30] The review aimed to identify potentially effective characteristics of interventions for this population to inform the development of a tailored ED-based intervention. The review identified 25 relevant primary studies, but found that quantitative evidence was not available to determine which, if any, characteristics may be most effective. However, the qualitative analysis of authors' experiences of testing the interventions highlighted a number of themes that may be important to consider when developing an ED-based intervention for people with terminal illness. These include flexibility of when, where and how to complete the intervention, clarity of instructions and provision of a safe environment for disclosure.

Building on the results of this scoping review, we developed an ED-based intervention in collaboration with key stakeholders named Let It Out (LIO). We then refined its design by incorporating insights from a trial we ran of an adapted version of the intervention tested in the general population during the COVID-19 pandemic,[31] and further stakeholder feedback. This study represents the next phase of the intervention development process.

## Aims and objectives

The primary objective of the study is to optimise the design of an ED-based intervention (LIO) tailored for people receiving palliative care from a UK hospice. The secondary objectives are to:

1. Assess whether the intervention and study procedures are acceptable and feasible in UK hospice services.

2. Explore the potential impact of the intervention on psychological and physical well-being, and identify any potential adverse effects.
3. Inform the choice of outcome measures and study design for possible future large-scale intervention evaluation.
4. Explore the causal mechanisms underlying potential intervention efficacy.

An exploratory objective is to explore the core concerns of people following referral to hospice care. This is one of the topics covered in the intervention expression sessions; responses could potentially inform strategies to minimise or address those concerns.

## METHODS AND ANALYSIS
### Design

A pragmatic, single-arm, multisite, mixed-methods longitudinal study. The study design has been developed in line with the latest Medical Research Council (MRC) guidelines,[32 33] which recommend an iterative intervention development and evaluation process. Mixed-methods, pragmatic study designs are recognised as an appropriate way for researchers to approach the development and evaluation of complex interventions in palliative care.[34] We are employing a convergent design to integrate quantitative measures of fidelity, feasibility, acceptability and possible impact, with semi-structured interviews and focus groups. In the qualitative aspect of the study, we will take a phenomenological approach to capture more nuanced feedback on the participants' experience of the intervention, including their views on its design, acceptability and any perceived effects.[34 35] The quantitative and qualitative data will be integrated through a process evaluation. As the focus of the study is on applying pragmatic mixed-methods to optimise intervention design, this study will not formally evaluate nor test *a priori* hypotheses regarding its efficacy or report p-values.

The study is approved by the University College London Research Ethics Committee (reference: 15281/002).

### Setting

The study will take place in six hospice services across England and Scotland. A list of participating sites is available from the authors.

### Participants

The study will be conducted in three population groups recruited from participating hospices:

#### Group 1

Up to 40 adults living with terminal illness receiving palliative care from a UK hospice (on an inpatient, day care, outpatient or community basis). A sample size of 40 is recognised by the National Institute of Health Research as being appropriate for intervention feasibility and pilot work that is not designed to evaluate efficacy.[36 37]

### Group 1 exclusion criteria

- ► Not expected to survive for longer than 10 weeks (based on clinical judgement).
- ► Lack of capacity to give informed consent or may be expected to lose capacity to give informed consent within 10 weeks (based on clinical judgement).
- ► Known diagnosis or history of severe depression, suicidal tendencies, psychotic symptoms, severe mood disorders, mania, schizophrenia, psychopathic/borderline personality disorder, severe and episodic (type 1) bipolar (affective) disorder (that is not well controlled).
- ► Not able to understand and communicate in English.

### Group 2

Up to 15 informal (adult friend or family) carers of Group 1 participants.

### Group 3

Up to 30 paid or voluntary hospice staff involved in organisation, delivery or implementation of psychological and/or emotional support.

### Recruitment
#### Group 1

We will employ convenience sampling. This is a pragmatic decision based on difficulties recruiting participants during the COVID-19 pandemic; the target numbers for recruited participants are in place to manage the scope of the research. We will assess recruitment rate as part of the study's feasibility assessment. The recruitment window has been selected based on research team capacity and funding availability.

Clinical and research staff at hospice sites will identify patients who may be eligible to participate between October 2020 and January 2021. They will make initial contact, explain the study and if the participant is interested, provide them with an invitation letter, email or link to the study website. The study website hosts the participant information sheet and a link to an online reply form. Participants can indicate on the reply form if they would or would not like to take part.

If participants select yes on the reply form, they can progress to the online informed consent and advance consent form, which asks what participants would like to happen to their data if they lose capacity over the course of their participation in the study. If they select no, they have the option to provide their reasons for not taking part. There is also an option to request further information from the research team before making a decision. If participants select this option, they are prompted to provide their contact details and a member of the research team will phone or email them to discuss the study further. If, having discussed the study, participants choose to take part, they can progress to the online informed consent and advance consent forms. The research team will inform the relevant hospice when a participant from their service consents to take part, so it

can be recorded on their patient notes. The participant will be reminded of their right to withdraw from the study at any time throughout the study in online materials and through direct communication with the research team.

In the initial phase of the study, all Group 1 participants will be invited to take part in a semi-structured interview. Depending on uptake, once a third (n=5) have been recruited to interview phase, we will assess the mix of participants in terms of demographic characteristics, to inform recruitment targets according to a purposive sampling framework.

#### Group 2

Informal carers will be recruited through researcher contact with Group 1 participants. If their carer indicates they may be interested in taking part in an interview, the research team will arrange a time to discuss the study; if interested in taking part, they will be sent a link to the online participant information sheet and informed consent form.

#### Group 3

Staff and volunteers will be recruited through liaison with clinical leads at each hospice site. The research team will approach eligible staff members to introduce the study, and if interested, provide them with a link to the online participant information sheet and informed consent form.

The study will be presented at hospice staff team meetings and internal newsletters to support recruitment.

### The intervention
#### Development

The LIO intervention has been developed in collaboration with a group of stakeholders, including psychologists, a psychiatrist, palliative care and hospice staff, palliative care specialist researchers and patient and public involvement (PPI) representatives. The intervention was first developed based on the findings of a scoping review of related interventions conducted by the research team.[29] Based on the review findings, the LIO intervention is flexible in terms of when, where and how it can be completed; provides clear, structured instructions; and emphasises the importance of completing sessions in a safe, comfortable setting. During the COVID-19 pandemic, the research team adapted the intervention to test its effect on distress in the general population during the pandemic (the Let It Out - COVID-19 response [LIO-C] study).[31] Based on feedback from participants in that study, adaptations were made to LIO to minimise any potential risk of causing short-term distress; namely, by incorporating clearer directions on how to express feelings with self-compassion.

#### Theoretical basis

The intervention development process has been guided by emotion regulation theory, which posits that the active component of ED-based interventions is the emotional arousal following expression of emotions.[38–40] This

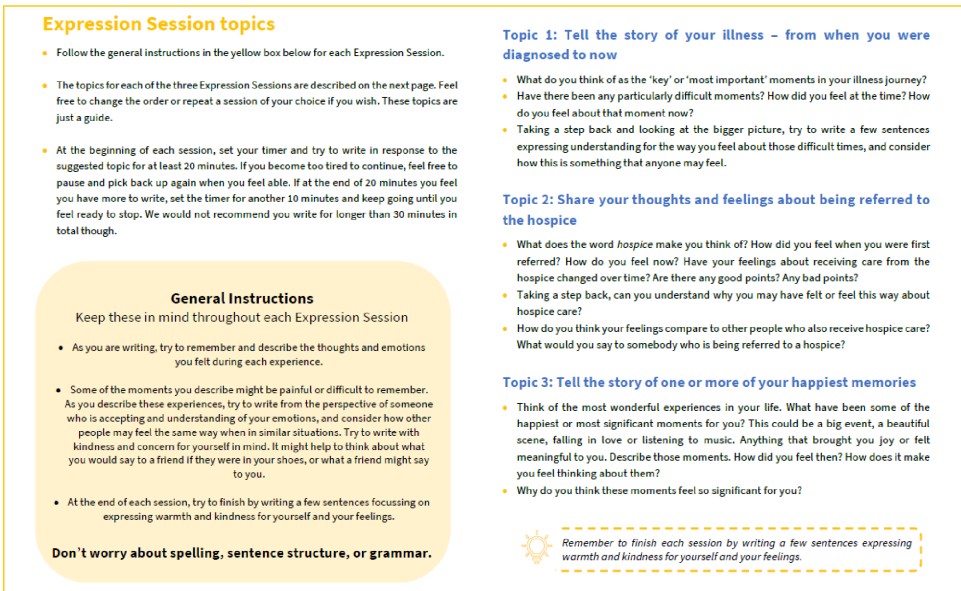

**Figure 1** General instructions and session specific prompts as part of Let It Out guide (written version).

account suggests that one is able to develop acceptance of their thoughts and feelings through mastery, self-efficacy and control of their emotions. This account also sits within theories underlying positive psychology and self-compassion interventions,[19 41] which propose that one is able to gain a better understanding of the emotions associated with their experiences by learning how to more helpfully structure thoughts and feelings. We recognise that there is unlikely to be a single process underlying ED; indeed, several alternative processes have been proposed to explain ED including emotional inhibition, cognitive adaptation and exposure.[42–45] One aim of this study, in line with the MRC complex intervention development guidance,[33] is to explore the mechanisms through which LIO may be effective.

### Outline

LIO is a self-guided intervention designed to enable people living with terminal illness to self-compassionately express their feelings in a way that may help to bring them comfort. In this study, participants will be sent a link to online intervention instructions, as well as a PDF intervention guide that can be downloaded from the online survey platform. The guide provides instructions for participants to work through the study and intervention. Over 2 weeks, participants are asked to express their feelings in a self-compassionate way in response to prompts in 3 separate, 20–30 minute 'expression sessions'. Figure 1 presents the general instructions given to participants to use across each session and the prompts for each session.

Participants are advised to complete sessions in a comfortable place, and, if possible, in private (although it is recognised this may not be feasible or wanted for all participants). Participants can handwrite, type or audio-record (using their personal mobile phone or laptop) their response to the prompts. There are 2 guides: one for those choosing to write, the other for those choosing

to audio-record. The full guides are available on reasonable request from the authors.

To make the intervention as accessible as possible, it is acceptable for participants to have support from a carer to help complete the expression sessions, and the associated questionnaires. This help can be delivered in any way that works best for the participant. The guide and website include signposts to other helpful resources that can support mental well-being. The guide also provides a space for participants to plan when and where they will complete each session, based on the theory of action planning to help bridge the intention-behaviour gap.[46]

### Data collection
#### Overview—group 1
Table 1 provides a detailed description of the participant timeline and outcome measures for Group 1 participants.

#### Expression sessions—group 1
Over the 2-week intervention period, participants will type, handwrite or audio-record (format chosen by the participant) their response to the three prompts. Handwritten and audio-recorded responses will be transcribed for analysis. It is not mandatory for participants to share their expression session responses if they would prefer to keep them private. Methods for data collection and transfer are summarised in table 2. Researchers will contact each participant by phone or email (participant preference) at the end of the first and second week of the intervention to check on their well-being. If during contact with the participant the research team suspects they may no longer have capacity for informed consent, or is experiencing severe distress, the hospice clinical team will be informed and capacity formally assessed. If the participant is deemed no longer eligible to take part in the study, they will be withdrawn and the directives from their advance consent form followed.

**Table 1** Group 1 participant timeline, outcome measures and analysis approach

| Measure | Time points administered | Description | Data type | Analysis approach | Relevant research objective* |
|---|---|---|---|---|---|
| Demographics | Baseline | Approximate date of hospice referral, age, gender, ethnicity, marital status, religious background, level of education, type of care (eg, inpatient, outpatient, day care), diagnosis, mental health history and COVID-19 status. | Quantitative | Descriptive statistics | 1. Feasibility and acceptability |
| Expression sessions | Three sessions over 2-week intervention period | Typed, handwritten or audio-recorded responses to three expression sessions that make up the LIO intervention. | Quantitative and qualitative | LIWC and BIR inductive thematic analysis (session 2) | 1. Feasibility and acceptability <br> 2. Underlying mechanisms <br> Exploratory objective (session 2) |
| Health-related outcome measures (see below for details) | Baseline, immediately after the final expression, and, 1 week, 4 weeks and 8 weeks after the final expression | See below for details. | Quantitative | Descriptive statistics and descriptive pre–post analysis | 1. Feasibility and acceptability <br> 2. Impact on well-being <br> 3. Outcome measure selection <br> 4. Underlying mechanisms |
| Integrated Palliative Care Outcome Scale (IPOS) – patient version[56] | | An updated, brief (10 questions, 17 items) version of the POS instrument. IPOS was developed specifically to capture a range of the most important concerns of people living with palliative-stage disease, covering symptoms, information needs, practical concerns, anxiety or low mood, family anxieties and overall feeling of being at peace. Items in question 2–9 are scored from 0 to 4, and overall scores can range from 0 to 68. Questions 1 and 10 are not scored. | | | |
| Patient Health Questionnaire 9[57] | | A 9-item instrument developed to screen for major depressive disorder, and validated in palliative care populations.[58] Items are scored 0 to 3, and it can be scored continuously or with validated cut-offs for mild (5 to 9), moderate (10 to 14), moderately severe (15 to 19) and severe depression (20 to 27).[59 60] The cut-off for a depression 'case' is 10. | | | |

Continued

**Table 1** Continued

| Measure | Time points administered | Description | Data type | Analysis approach | Relevant research objective* |
|---|---|---|---|---|---|
| Generalised Anxiety Disorder 7[61] | | A validated 7-item anxiety screening instrument that is used to screen for anxiety in palliative care settings. Items are scored 0 to 3, and total scores range from 0 to 21. Severity of anxiety symptoms can be classed as none (0 to 4), mild (5 to 9), moderate (10 to 14) or severe (15 to 21). The cut-off score for a generalised anxiety disorder 'case' is 10. | | | |
| Single-item Sleep Quality Scale[62] | | A short, pragmatic, tool for assessing sleep quality that has been validated in people with insomnia and depression, selected to reduce burden on participants from lengthier alternatives such as the Pittsburgh Sleep Quality Index.[63] A single Visual Analogue Scale asks respondents to rate their overall quality of sleep over the last 7 days on a scale of 0 to 10. Scores are used to categorise sleep quality as: terrible (0), poor (1 to 3), fair (4 to 6), good (7 to 9) and excellent (10). | | | |
| Uptake of existing mental health services | | A 3-item form developed by our research team specifically for this study asking participants if they have ever (a) sought help from a healthcare professional for their mental health; (b) sought help from a healthcare professional for their mental health over the past 2 weeks; and (c) any plans to seek help from a healthcare professional for their mental health in the future. Response options are 'yes', 'no' or 'prefer not to say'. | | | |
| Meaningfulness/ personal/mood ratings | Immediately after each expression session | A 3-item, 7-point Likert scale based on a measure used in a study testing a similar intervention,[64] measuring how personal and meaningful participants' felt their expression session was (not very personal/not very meaningful (−3) to (+3) very personal/very meaningful). The last item will be a mood check where participants are asked to rate how they feel on a 7-point scale (very sad (−3) to (+3) very happy). | Quantitative | Descriptive statistics | 1. Feasibility and acceptability<br>2. Impact on well-being<br>3. Outcome measure selection<br>4. Underlying mechanisms |

Continued

**Table 1** Continued

| Measure | Time points administered | Description | Data type | Analysis approach | Relevant research objective* |
|---|---|---|---|---|---|
| Feedback form | Immediately after final expression session | Participants will be asked to complete an online feedback form exploring their experience of completing the intervention, if they needed any assistance from a carer and the nature of that assistance, and their fidelity to the protocol. The feedback form questions were informed by work on ethical research in vulnerable populations[65] and the theoretical framework of acceptability for healthcare system interventions.[53] | Quantitative and qualitative | Descriptive statistics and combined deductive and inductive thematic analysis | 1. Feasibility and acceptability<br>2. Impact on well-being<br>3. Outcome measure selection<br>4. Underlying mechanisms |
| Semi-structured interview | Within 8 weeks of final expression session | Exploring participants' experience of the intervention and questionnaires, including whether they experienced any negative feelings during or after each expression, whether they found the intervention helpful and the appropriateness and relatability of the expression session prompts, expanding on the topics covered in the feedback form | Qualitative | Combined deductive and inductive thematic analysis | 1. Feasibility and acceptability<br>2. Impact on well-being<br>3. Outcome measure selection<br>4. Underlying mechanisms |

*All measures will contribute to the primary objective: optimisation of the intervention as informed by a process evaluation (see table 3).
BIR, blinded independent review; LIO, let it out; LIWC, linguistic inquiry word count.

**Table 2** Methods of expression session data collection and transfer for Group 1 participants

| Method of disclosure | Data collection | Method for transfer to research team |
|---|---|---|
| Typed into platform | Participant types directly into text box on online survey platform (Research Electronic Data Capture [REDCap] via Data Safe Haven). | Research team have access to data entered onto REDCap via Data Safe Haven. |
| Typed on device | Participant types into word processor on own device. | Participant saves file on own device and uploads it to the REDCap via Data Safe Haven survey platform. |
| Handwritten | Participant handwrites responses on own paper. | Participant takes photo of handwritten responses using mobile phone camera and sends via WhatsApp to research team* (end-to-end encryption). Photos transferred to Data Safe Haven and deleted from phone. |
| Audio-recorded on smart phone | Participant records themselves using either inbuilt recording/notes application on their phone or the Rev application (free to download). | Participant sends audio-recording via WhatsApp to research team* (end-to-end encryption). Audio-recording transferred to Data Safe Haven and deleted from phone. |
| Audio-recorded on laptop/computer | Participant records themselves using free Audacity software. | Participant saves file on own device and uploads it to the REDCap via Data Safe Haven survey platform. |

*The research team will have a dedicated study phone that a member of the research team keeps only on their person during working hours (09:00–17:00 Monday–Friday) and is kept in a secure locked cabinet outside of those hours. The phone will be checked once a day for recordings or photos received, at which point the recording will be transferred to the dedicated project file on Data Safe Haven and deleted from the phone.

### Self-report measures—group 1

All self-report data is collected online using the Research Electronic Data Capture (REDCap) online survey platform.[47 48] Group 1 participants will complete a demographics questionnaire at baseline (prior to receiving the intervention). They will also complete a series of health-related questionnaires at baseline and 1, 4 and 8 weeks after the final intervention session (secondary outcome measures). Group 1 participants will also be prompted to log in to the platform during or after each expression session to indicate they have completed the session, and (if they wish) to share their response with the research team. Directly after each expression session they will also be prompted to complete a short self-report questionnaire to monitor their mood, and perception of how meaningful and personal the session was (secondary outcome measures). After the final expression session, participants will also complete a feedback form (primary outcome measure).

### Semi-structured individual interviews (primary outcome measure)—group 1

Semi-structured interviews (n≤15) will be conducted within 8 weeks of the participant's final expression session. Interviews will be conducted by DM, following a topic guide to explore participants' experience of the intervention and questionnaires, including whether they experienced any negative feelings during or after each expression, whether they found the intervention helpful and the appropriateness and relatability of the expression session prompts, expanding on the topics covered in the feedback form.

Interviews will take place virtually and will be audio-recorded using Microsoft Teams, to minimise face-to-face interactions in line with government COVID-19 guidance. Recordings will be transcribed verbatim for analysis; transcripts will be pseudonymised and personally identifiable information redacted from the transcript.

### Overview—group 2 and 3 (primary outcome measures)

Semi-structured interviews will also be conducted with ≤15 informal carers of Group 1 participants after they have completed their final expression session. Focus groups or semi-structured interviews (depending on practicalities of arranging a suitable time) with ≤30 hospice staff and volunteers will also be conducted. As with Group 1, interviews and focus groups will be conducted by DM, and will follow topic guides designed to explore participants' views on the design and content of LIO; if and how it may need to be adapted; potential risks of harm or benefit; where LIO might fit into current pathways of care; and what help may be needed to support its implementation. All Group 2 and 3 participants will complete an online demographics questionnaire prior to taking part in the interview/focus group.

As with Group 1, interviews and focus groups will take place virtually and will be audio-recorded using Microsoft Teams. Recordings will be transcribed verbatim, pseudonymised and personally identifiable information redacted from the transcript for analysis.

### Data analysis
#### Primary objective: process evaluation and intervention proposal

Using the qualitative and quantitative data collected from Groups 1, 2 and 3 (and in combination with evidence that has informed this research protocol from prior work (reviews and a trial) conducted by the research team), a triangulation approach will be applied drawing on

process evaluation methodology.[35] Process evaluation is recognised as a valuable, pragmatic method to optimise intervention design and inform evaluation during feasibility testing phases.[35] Through the process evaluation, we aim to develop a preliminary theoretical model to inform understanding of potential causal mechanisms of LIO and any contextual and implementation factors that may be associated with any potential variation in outcomes. Insights from the process evaluation will inform the final intervention proposal. Table 3 describes the framework that will be used to guide the integration of mixed-methods data for the process evaluation. We developed the framework based on the MRC guidance on conducting process evaluations.[35]

Analysis will be led by DM in iterative consultation with the research team and advisory group. Codes, themes, descriptive statistics, data integration and conclusions generated by DM will be checked at key stages throughout the project by the research team. This will help minimise any risk of a positivity bias arising from DM's close connection to the intervention being developed, and help validate the integrity of DM's interpretation of the data. Thematic analysis will be carried out using QSR International NVivo V.11.4 software.[49] Quantitative responses will be summarised using SPSS V.24.[50]

## Secondary objectives
### Feasibility and acceptability
Due to the preliminary nature of this study, no prespecified criteria have been defined to establish acceptability of the LIO intervention, or the feasibility of the study procedures. The following measures of feasibility and acceptability will be analysed

1. Recruitment and retention
   a. Descriptive statistics of recruited participant numbers and demographics.
   b. Percentage of participants recruited completing each data collection time point.
2. Adherence
   a. Percentage of participants completing all three expression sessions (a) within 20–30 min time frame and (b) in a private space.
   b. Blinded independent assessor review of texts to determine which prompt was being responded to.
   c. Linguistic inquiry word count (LIWC) analysis of expression session texts/transcripts[51] to analyse percentages of emotion, self and prompt-related words. LIWC is a computer program that calculates word count, and percentages of words used that reflect different emotions, thinking styles and concerns.
3. Acceptability
   a. Descriptive quantitative analysis of responses to the feedback form.
   b. Combined deductive and inductive qualitative thematic analysis of free-text responses to feedback form and interviews with patient, carer and staff participants.[52] Data will be coded to the seven constructs of the theoretical framework of acceptability

for healthcare system interventions (deductive component).[53] Within each construct, further codes will be developed from the raw data (inductive component). DM will lead the application and development of codes. These will then be checked and adjusted by the research team (intercoder review and code testing), and data re-coded. This process will be repeated iteratively until code/theme saturation is reached.[54]

### Impact on psychological and physical well-being
a. Inductive thematic analysis of interviews and feedback from free-text responses from patients, carer and staff participants. DM will develop codes from the raw data, to be checked and adjusted by the research team; data will be re-coded, checked and adjusted until code saturation is reached.
b. Descriptive pre–post analysis of changes in outcome measures (Integrated Palliative Care Outcome Scale, Patient Health Questionnaire 9, Generalised Anxiety Disorder 7, Sleep Quality Scale, uptake on mental health services): mean and standard deviation of scores at each time point.
c. Descriptive summary of number of adverse effects reported by clinical team and participants.

### Choice of outcome measure
Inferences of potentially suitable outcome measures will be based on results from descriptive pre–post analysis of outcome measures, and qualitative analysis of interviews and feedback from free-text responses from patients, carers and staff participants regarding reported impact of the intervention.

### Potential underlying mechanisms
Responses to expression session prompts will be analysed quantitatively using LIWC software[51] and triangulated with indicators of intervention impact on well-being to inform development of a preliminary theoretical model.

### Core concerns on referral to hospice
Responses to expression session prompts will be analysed using inductive thematic analysis. DM will develop codes from the raw data, to be checked and adjusted by the research team; data will be re-coded, checked and adjusted until code saturation is reached.

## PPI
The funding application for this programme of work was reviewed by Mr Peter Buckle, a Marie Curie PPI representative. Peter is also a member of the advisory group overseeing the conduct of the PhD studentship of which this study is a part. Peter, along with a second PPI representative (Dori-Anne Finlay) have reviewed the study and intervention design. Peter and Dori-Anne will be consulted at key stages of the research. Furthermore, the study as a whole has been designed based on the principles of co-design; the data from participants will be used to inform the design of the final intervention and potential future research.

**Table 3** Framework guiding analysis and integration of mixed-methods data for process evaluation

| | Research questions | Data source | Data analysis |
|---|---|---|---|
| Implementation | Coverage and reach (retention):<br>▲ What percentage of Group 1 participants completed the intervention?<br>▲ What are the demographics of participants who took part in the study and completed the intervention?<br>▲ What were the reasons given for non-participation? | Recruitment/data collection records<br>Demographics questionnaire<br>Researcher reflective notes and informal feedback from research sites; withdrawal questionnaire | Descriptive statistics<br>Inductive thematic analysis |
| | Fidelity, frequency and duration (adherence):<br>▲ To what extent was the intervention implemented as planned?<br>— What percentage of participants completed all three expression sessions, within the 20–30 min time frame, in a private space?<br>— What percentage of expression sessions were assigned to the correct prompt by a blinded assessor?<br>— To what extent do expression session transcripts use emotion and positive emotion words? | Recruitment/data collection records<br>Group 1 feedback form<br>Interviews and focus groups with Group 1, 2 and 3<br>Expression session texts | Descriptive statistics<br>Blinded assessor review<br>LIWC analysis |
| | Acceptability:<br>▲ How acceptable is the intervention for people living with terminal illness receiving care from hospices, their informal carers and hospice staff?<br>▲ What factors affect its acceptability? | Group 1 feedback form<br>Interviews and focus groups with Group 1, 2 and 3 | Descriptive analysis and inductive thematic analysis coded to concepts in the TFA |
| Impact and mechanisms | Impact:<br>▲ What are the perceived benefits of the intervention from the perspective of people living with terminal illness, their informal carers and hospice staff?<br>▲ Are there any changes in scores of psychological and physical health measures before and after (immediately, 1, 4 and 8 weeks post intervention)? | Group 1 feedback form<br>Interviews and focus groups with Group 1, 2 and 3<br>Physical and psychological health-related outcome measures and mood checks | Inductive thematic analysis<br>Descriptive pre-post statistics |
| | Mechanisms of impact:<br>▲ What cognitive processes are taking place during each expression session?<br>▲ Are there any patterns in the characteristics of participants who perceive benefits? | Expression session texts<br>Demographic questionnaires | LIWC<br>Descriptive statistics |
| Contextual factors | What factors at political, economic, organisational and individual level affected, or might be expected to affect, the implementation and/or impact?<br>What options are there to minimise any barriers to successful implementation? | Group 1 feedback form<br>Interviews and focus groups with Group 1, 2 and 3<br>Researcher reflective notes and informal feedback from research sites; withdrawal questionnaire | Inductive thematic analysis |

LIWC, linguistic inquiry word count; TFA, the theoretical framework of acceptability.

## Data management

All survey-based data (demographics and health-related questionnaires) will be collected using REDCap[47 48] via University College London Data Safe Haven. The Data Safe Haven has been certified to the ISO27001 information security standard and conforms to National Health Service Digital's Information Governance Toolkit. Built using a walled garden approach, the data is stored, processed and managed within the security of the system, avoiding the complexity of assured endpoint encryption. A file transfer mechanism enables information to be transferred into the walled garden simply and securely. A data management plan has been registered and approved by the University College London Data Protection Office outlining methods to maintain participant privacy and data integrity and confidentiality. At the end of the project, anonymised data will be uploaded to open-access data repository ReShare.[55]

## Ethics and dissemination
### Confidentiality

All participant data will be collected via the secure REDCap via Data Safe Haven platform,[47 48] or Microsoft Teams (interviews/focus groups), except audio-recordings or photographs of expression sessions, which will be sent directly to a dedicated study telephone via WhatsApp secured by end-to-end encryption. The phone will be carried by a member of the research team or stored in a locked drawer. Only members of the research team will have access to participant data. Consent and data collection forms will be securely stored in the University College London Data Safe Haven. During analysis, data will be pseudonymised using a unique ID that could not be linked to a participant's identity by anyone outside of the research team. Data will be anonymised 3 months after participants' involvement in the study ends.

## Serious adverse events

Any serious adverse events (SAEs) attributed to a person's participation in the study will be recorded on the online study platform, and in the participant's medical records. SAE forms will be provided by the chief investigator to the sponsor within 5 days of becoming aware of the event.

## Dissemination

The findings of this study will be disseminated through peer-reviewed scientific journals, conferences and traditional, online and social media.

**Acknowledgements** The LIO study team are grateful to the group of advisors who have provided invaluable feedback on the design of the LIO intervention and study design to date: Peter Buckle, Jason Davidson, Dori-Anne Finlay, Dr Syd Hiskey, Dr Joana Johnson, Professor Marc Serfaty, Dr Nick Troop. We are also grateful to the research and clinical leads and supporting staff at participating sites for their feedback on the intervention and study design, and support through the research governance process, particularly: Dr Anne Finucane, Dr Emma Carduff, Dr John MacArtney, Dr Amara Nwosu, Dr Adrian Tookman, Gail Holloway, Rachel Perry.

Thank you also to Dr Sabine Best and Dr Briony Hudson for support in facilitating multisite research across Marie Curie hospice sites.

**Contributors** The study and intervention design was led by DM. NK, BC and PS have collaborated at all stages on study and intervention development. NK is the chief investigator, based at University College London. DM is the PhD student managing the project. DM led the writing of this manuscript. All authors contributed to and approved the final draft.

**Funding** This work was supported by Marie Curie (MCCC-FCH-18-U) and Economic and Social Research Council collaborative funding (ES/P000592/1). This work was also supported by core funding by Marie Curie (grant number: MCCCFPO-16-U) and an Alzheimer's Society (AS) funded fellowship awarded to NK (grant number: 399 AS-JF-17b-016).

**Competing interests** None declared.

**Patient and public involvement** Patients and/or the public were involved in the design, or conduct, or reporting, or dissemination plans of this research. Refer to the Methods section for further details.

**Patient consent for publication** Not required.

**Provenance and peer review** Not commissioned; externally peer reviewed.

**ORCID iD**
Daisy McInnerney http://orcid.org/0000-0002-8921-2215

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
