## [Reviewer comments · BMJ Open]

ARTICLE DETAILS

TITLE (PROVISIONAL)	Let It Out (LIO) study: protocol for a mixed-methods study to optimise the design and assess the feasibility of an online emotional disclosure-based intervention in UK hospices
AUTHORS	McInerney, Daisy; Candy, Bridget; Stone, Patrick; Kupeli, Nuriye

VERSION 1 – REVIEW

REVIEWER	Bennett, Charles South Carolina College of Pharmacy/USC Campus
REVIEW RETURNED	14-Dec-2020

GENERAL COMMENTS	This is a strong protocol. It is interesting to read. One could consider a table highlighting how the data are cross walked with the theoretic construct for the study.
---

REVIEWER	Steindal, Simen Lovisenberg Diaconal University College
REVIEW RETURNED	09-Feb-2021

GENERAL COMMENTS	An interesting and mostly well written protocol P4, L30 immediately after – after what? Please clarify P4 Method – please state that you employ a mixed methods design. Please describe shortly how quantitative and qualitative data will be integrated since you have a mixed methods design. The scientific rationale for conducting the study is well described. The primary objectives and secondary objectives 1-4 are clear. However, please clarify the relevance of secondary objective 5. P 5 Do you have a sequential or convergent mixed methods design? Will quantitative data inform the qualitative data collection? P7, L48 “qualitative interviews” please replace with “semi-structured individual interviews”. Several places you use the less than or equal to sign regarding number of hospice and number of participants.
---

	Could this be revised to avoid confusion? P9, L 42 what does PPI representatives mean? Please spell fully out first time you use. Recruitment of group 1 for the intervention, convenience or consecutive or other type of sampling strategy? Recruitment of group 1 participants for interviews, purposeful or convenience or other type of sampling strategy? Please clarify. P 14, L49 SQS validated in palliative care populations? P 14, L58, Uptake of existing mental health service, has this form been validated? P 14-15 Uptake of existing mental health services – has this been validated in this population? P15 please describe shortly the topics you will explore in the interviews with group 2 and 3 P15, L50 will the recording be transcribed verbatim? P16-17 Please elaborate more clearly on how you will integrate quantitative and qualitative data in data analysis and the results. This is essential for a mixed methods design.
--	--

REVIEWER	Lawton, Andrew Dana-Farber Cancer Institute
REVIEW RETURNED	11-Feb-2021

GENERAL COMMENTS	Major Comments:  - The authors describe the development and optimization of an online, self-guided emotional disclosure-based intervention, called Let It Out (LIO), tailored for people with terminal illness receiving hospice care. This manuscript describes their mixed-methods study protocol, which comes in advance of eventual evaluation of the intervention’s efficacy. The relevance of the LIO intervention is highlighted by the current COVID era, when telehealth and other remote interventions are especially critical, and may extend beyond this period as well. - In this well-written manuscript, the authors describe the gap in the literature regarding emotional disclosure interventions and how their protocol is an important next step in filling that gap. - I found the manuscript and protocol to be of good quality, clear, well-referenced, and worthy of publication. - I found it particularly valuable that the authors built-in several methods for monitoring participants’ mood and wellbeing while engaging with the ED intervention, including a self-report mood questionnaire after each expression session and a phone call/email in week 2 from the research team to check-in on wellbeing. This is, of course, particularly important given the sensitive nature of
--

	the intervention's questions, prompts, and content.
--	---

VERSION 1 – AUTHOR RESPONSE

First reviewer comments

This is a strong protocol. It is interesting to read.

One could consider a table highlighting how the data are cross walked with the theoretic construct for the study.

Thank you for your positive feedback on the protocol – we are glad you found it of interest. In response to your comment regarding linking the theoretic construct with the data being collected (and to some of the feedback from reviewer 2, and the editors) we have developed Table 3. The aim of this table is to provide an at-a-glance, visual summary of how in this study, guided by a pragmatic research philosophy, we plan to integrate multimodal data to inform the optimisation of the intervention, such that it is tailored for practical implementation in a real-world setting.

Second reviewer comments

An interesting and mostly well written protocol

Thank you for this feedback. We appreciate your detailed and thoughtful critique and have tried to address each point. Your guidance has certainly resulted in a stronger, more coherent protocol.

P4, L30 immediately after – after what? Please clarify

We have changed the phrasing to clarify we mean immediately after the last intervention session (Line 31).

P4 Method – please state that you employ a mixed methods design. Please describe shortly how quantitative and qualitative data will be integrated since you have a mixed methods design.

We have included a statement that this is a mixed methods design in the title, Methods section of the abstract, and in the Design subsection of the Methods and Analysis section. In the abstract, we have noted that the mixed-methods data will be integrated using process evaluation to triangulate the data, and provided more detail on this integration method in the Methods section (Lines 115-121, 372 – 387, Table 3).

The scientific rationale for conducting the study is well described. The primary objectives and secondary objectives 1-4 are clear. However, please clarify the relevance of secondary objective 5.

Thank you for this feedback. We agree the 5th objective (exploring the core concerns of people following referral to hospice care) is not as clearly aligned with the primary objective and other secondary objectives of the study. However, since data from the Expression Sessions could provide important insights into what the main fears and concerns are for patients on referral to hospice services (and therefore, potentially inform strategies of minimising or addressing those fears and concerns), we argue it is worthy of inclusion and analysis. In recognition of your valid point, we have reframed objective 5 as an Exploratory objective and added a brief explanatory sentence to explain its relevance (Line 106 - 108).

P 5 Do you have a sequential or convergent mixed methods design? Will quantitative data inform the qualitative data collection?

Thank you for this request for further clarity. We are employing a convergent design; quantitative data will not inform the qualitative data collection. The feedback form (which comprises both quantitative multiple choice and qualitative open-ended questions) is standardised and administered immediately after the participant completes the final expression session. The semi-structured

interviews and focus groups are guided using a topic guide, and not directly informed the quantitative data provided by participants.

We have updated the manuscript to clarify this (Lines 115 – 123).

P7, L48 “qualitative interviews” please replace with “semi-structured individual interviews”.

Thank you for noticing this. We have replaced the term as advised (Line 117).

Several places you use the less than or equal to sign regarding number of hospice and number of participants. Could this be revised to avoid confusion?

Thank you for raising this. We have considered your suggestion, and have now provided a specific number of sites (Line 126). However, we would like to continue using the less than or equal to sign with regards to the number of participants. We have justified our decision to do so in Line 148 – 151 of the manuscript; this is a pragmatic decision made on the basis of difficulties recruiting participants during the COVID-19 pandemic. In line with this pragmatic philosophy, the ‘maximum’ numbers for recruited participants are in place to manage the scope of the research. We will assess recruitment rate as part of the objective of this study to assess feasibility; as such, it may be that fewer than the targeted number of participants are recruited within the recruitment window for the study. That said, where the symbol has been used at the start of a sentence, we have replaced it with the phrasing ‘Up to’ to reduce any confusion.

P9, L 42 what does PPI representatives mean? Please spell fully out first time you use.

Thank you for spotting this. We have spelt this out as Patient and Public Involvement (PPI) (Line 190).

Recruitment of group 1 for the intervention, convenience or consecutive or other type of sampling strategy? Recruitment of group 1 participants for interviews, purposeful or convenience or other type of sampling strategy? Please clarify.

For Group 1 recruitment, we will employ convenience sampling as a pragmatic decision in light of the difficulties opening sites and relative unknowns of recruiting during the COVID-19 pandemic (Line 148). In the initial phase of the study, all Group 1 participants will be invited to take part in the study interview; after one third (n=5) have been recruited, we will assess the mix of participants in terms of demographic characteristics, to inform recruitment targets according to a pre-defined purposive sampling framework (Line 172- 175).

P 14, L49 SQS validated in palliative care populations?

This is a relatively newly developed measure (published in 2018) and to our knowledge, it has not been validated in palliative care populations. We chose it specifically to minimise any burden that could be associated with the more widely validated, but significantly longer, questionnaires such as the Pittsburgh Sleep Quality Index. We were particularly cautious as this is just one of several measures being administered, and we therefore did not want to over-burden participants.

P 14, L58, Uptake of existing mental health service, has this form been validated?

No, this is a form that was developed specifically for this study to provide insight into whether people have, or intend to, request help from mental health services.

This is clarified in Table 1 in the description of the measure.

P 14-15 Uptake of existing mental health services – has this been validated in this population?

As described above, no, this is not a validated measure, but a form developed specifically for this study by the research team.

P15 please describe shortly the topics you will explore in the interviews with group 2 and 3

We have added a brief overview of the topics covered on 361 - 365 (including views on the intervention content and design, potential risks or benefits, where the intervention could fit into current pathways of care and what may be needed to support its implementation).

P15, L50 will the recording be transcribed verbatim?

Yes, the recording will be transcribed verbatim. This is now stated on Lines 282 and 368.

P16-17 Please elaborate more clearly on how you will integrate quantitative and qualitative data in data analysis and the results. This is essential for a mixed methods design.

In the original version of this manuscript that we first submitted, we stated that quantitative and qualitative data would be triangulated/integrated through process evaluation, and that we would use the findings of the process evaluation to inform the optimisation of the intervention. To clarify this integration process, we have provided a summary table (Table 3) describing how each type of data will be analysed in accordance with our process evaluation framework. We created the framework based on the examples provided in the Medical Research Council (MRC) guidance on conducting process evaluations. We hope this provides sufficient clarity on how the data will be integrated.

Third reviewer comments

The authors describe the development and optimization of an online, self-guided emotional disclosure-based intervention, called Let It Out (LIO), tailored for people with terminal illness receiving hospice care. This manuscript describes their mixed-methods study protocol, which comes in advance of eventual evaluation of the intervention's efficacy. The relevance of the LIO intervention is highlighted by the current COVID era, when telehealth and other remote interventions are especially critical, and may extend beyond this period as well.

In this well-written manuscript, the authors describe the gap in the literature regarding emotional disclosure interventions and how their protocol is an important next step in filling that gap.

I found the manuscript and protocol to be of good quality, clear, well-referenced, and worthy of publication.

I found it particularly valuable that the authors built-in several methods for monitoring participants' mood and wellbeing while engaging with the ED intervention, including a self-report mood questionnaire after each expression session and a phone call/email in week 2 from the research team to check-in on wellbeing. This is, of course, particularly important given the sensitive nature of the intervention's questions, prompts, and content.

Thank you very much for your positive feedback; it is really encouraging that you can see the value in our work, and appreciate the thought that has gone into designing the built-in methods to monitor participants' mood.

VERSION 2 – REVIEW

REVIEWER	Steindal, Simen Lovisenberg Diaconal University College
REVIEW RETURNED	29-Mar-2021
GENERAL COMMENTS	The authors have responded to the reviewers' comments/suggestions. The manuscript is improved.